# Change in Liver Fibrosis Associates with Progress of Diabetic Nephropathy in Patients with Nonalcoholic Fatty Liver Disease

**DOI:** 10.3390/nu15143248

**Published:** 2023-07-22

**Authors:** Yoshiko Terasaka, Hirokazu Takahashi, Kazushi Amano, Koshiro Fujisaki, Shotaro Kita, Kaori Kato, Koujin Nakayama, Yuko Yamashita, Shuji Nakamura, Keizo Anzai

**Affiliations:** 1Department of Internal Medicine, Division of Metabolism and Endocrinology, Faculty of Medicine, Saga University, Saga 849-8501, Japan; 2Internal Medicine, Heiwadai Hospital, Miyazaki 880-0034, Japanh-kouwakai@heiwadai.or.jp (Y.Y.);; 3Liver Center, Saga University Hospital, Faculty of Medicine, Saga University, Saga 849-8501, Japan; 4Fujisaki Clinic, Kagoshima 891-0141, Japan; 5Ryutokukai Medical Corp, Tsuruta Hospital, Miyazaki 881-0016, Japan

**Keywords:** prognosis, renal outcome, diabetes complications, NASH, multidisciplinary care

## Abstract

Diabetic nephropathy (DN) is a major complication of diabetes. Nonalcoholic fatty liver disease (NAFLD) is common in diabetes, and liver fibrosis is a prognostic risk factor for NAFLD. The interaction between DN and liver fibrosis in NAFLD remains unclear. In 189 patients with DN and NAFLD who received an education course about diabetic nephropathy, liver fibrosis was evaluated using the fibrosis-4 (FIB-4) index. The association between the outcome of DN and changes in liver fibrosis was examined. The FIB-4 index was maintained at the baseline level in patients with improved DN, while it was increased in other patients. The ΔFIB-4 index was positively correlated with changes in albuminuria and proteinuria (ρ = 0.22, *p* = 0.004). In a multivariate analysis, changes in albuminuria and proteinuria were associated with the ΔFIB-4 index (*p* = 0.002). Patients with a progressive FIB-4 index category from baseline to 5 years showed a lower event-free survival rate after 5 years than patients with an improved FIB-4 index category (*p* = 0.037). The outcome of DN is associated with changes in liver fibrosis in patients with diabetes, NAFLD and DN. Developing a preventive and therapeutic approach for these conditions is required.

## 1. Introduction

Diabetic nephropathy (DN) is one of the vascular complications of diabetes, the progression of which causes renal failure leading to the need for dialysis and kidney transplantation [1,2,3,4]. The burden of dialysis and kidney transplantation for a large number of patients with diabetes encompasses medical, epidemiological and economic issues [5,6]. To prevent the occurrence and aggravation of DN, accumulating evidence has indicated that comprehensive care, not just treatment for hyperglycemia, but also treatment for hypertension, dyslipidemia and obesity, is important [7,8]. Multidisciplinary care and education for these patients contributes to reducing the progression of risk, improving patients’ adherence to treatment and delaying the initiation of dialysis [9,10,11].

Nonalcoholic fatty liver disease (NAFLD) is a common chronic liver disease mainly related to insulin resistance and obesity [12]. In type 2 diabetes, the prevalence of NAFLD is estimated to be 55.5% [13]. Glycemic control for diabetes is associated with the disease course and prognosis of NAFLD, including hepatic fibrosis, liver cancer and mortality [14,15]. Recent epidemiological studies and meta-analyses have shown that hepatic fibrosis is the most significant risk factor for overall mortality, including cardiovascular disease and for liver-related events such as liver failure, liver cancer and liver transplantation [16,17,18]. Body weight reduction and pharmacological therapy for diabetes such as glucagon-like peptide-1 (GLP-1) agonists and sodium glucose cotransporter 2 (SGLT2) inhibitors ameliorate pathological findings of NAFLD [19]. These studies have suggested that comprehensive care for diabetes and glycemic control are important in patients with diabetes and NAFLD to reduce the risk of cardiovascular disease, liver-related events and mortality.

NAFLD, DN and type 2 diabetes have a common background and risks, such as obesity and hypertension [20,21,22]. Therefore, we hypothesize that there is an interaction between changes in liver fibrosis in NAFLD and renal outcome in DN. Therefore, we conducted a single-center, retrospective, observational study to investigate the effect of multidisciplinary care and education of patients with DN on liver fibrosis in NAFLD. We also aimed to investigate the association between renal and hepatic outcomes in patients with diabetes, DN and NAFLD.

## 2. Methods

### 2.1. Study Design and Patients

All the patients with diabetes and DN who visited Heiwadai Hospital from September 2013 to December 2015 were consecutively recruited to an education course for diabetic nephropathy. A total of 263 patients were enrolled in the education course and included in this study. Medical records of the patients were retrospectively reviewed. Annual clinical data for up to 5 years were collected from 207 patients, and 56 patients were excluded owing to missing data or a concomitant condition (viral hepatitis, habitual alcohol consumption >30 g/day in men and >20 g/day in women, and hyperthyroidism) (Appendix A). No patients had a diagnosis of liver cancer, autoimmune liver disease, drug-induced hepatotoxicity, hemochromatosis, or Wilson’s disease. No patients had hematological disease or kidney disease, such as chronic glomerulonephritis, vasculitis, polycystic kidney disease, or renal cancer. NAFLD was defined according to the Hepatic Steatosis Index (HSI) [23]. The HSI was calculated using the following formula: HSI = 8 × alanine aminotransferase (ALT)/aspartate aminotransferase (AST) ratio + body mass index (BMI) (+2 for diabetes; +2 for female sex). Patients with an HSI > 30 were considered to have NAFLD [23], and 189 patients were finally analyzed (Appendix A). Information on renal prognostic outcome (renal failure defined as estimated glomerular filtration rate [eGFR] < 15 mL/min/1.73 m^2^ and/or initiation of dialysis) was also collected after 5 years. The study protocol was approved by the Clinical Research Ethics Review Committee of Heiwadai Hospital and Saga University Hospital in accordance with the principles of the 1975 Declaration of Helsinki, revised in 2013. An opt-out procedure was used for patients to withdraw consent in response to inclusion in the study.

### 2.2. Education Course for Diabetic Nephropathy

All patients enrolled in the present study were involved in the initial educational course for DN at Heiwadai Hospital. This course consisted of three 90-minute sessions. A physician, a pharmacist, a medical technologist, and a registered nutritionist facilitated the course and conducted lectures for patients and their families to teach the pathogenesis of diabetes and hypertension as well as related medication and examinations. Lectures also explained the pathogenesis of DN and renal prognosis. Therapeutic goals for glycemia, blood pressure, and cholesterol were indicated in the lectures. Information on the standard and ideal diet, including total calories, nutrient components, and the importance of a low salt diet (6 g/day), were taught. To better understand the low salt diet, Japanese-style miso soup was tasted in class and a quiz session using food samples was presented. Patients with an albumin (U-Alb)/urine creatinine (U-Cre) ratio ≥ 300 mg/g or urine protein (U-Pro)/U-Cre ratio ≥ 0.5 mg/g were instructed to avoid an overload of protein. All patients were instructed to stop smoking. After the initial educational course for DN, individual education was continued as required.

### 2.3. Physical Examination and Biochemical Measurements

Venous blood samples and urine samples were obtained. Hemoglobin A1c (HbA1c) values, platelet count, creatinine concentrations, AST concentrations, and ALT concentrations were measured in the blood samples. U-Alb, U-Pro, and U-Cre concentrations were measured in the spot urine samples using conventional laboratory techniques. During the observation period of 5 years, creatinine, HbA1c, and salt intake were measured every 6 months. U-Alb, U-Pro, and U-Cre were annually measured. Body weight, BMI, blood pressure, platelet count, AST, and ALT were annually recorded from baseline to 3 years and at 5 years. The eGFR was calculated using the following equation:  194  ×  creatinine (mg/dL)^−1.094^  ×  age (years)^−0.287^ for men; the values obtained for women were multiplied by 0.739 [24]. Salt intake was estimated with spot urine samples using Tanaka’s equation [25].

The fibrosis-4 index (FIB-4 index) was calculated as follows: age × AST/platelets × [(ALT)^1/2^]. Risk categories for advanced liver fibrosis were evaluated as follows: an FIB-4 index <1.3 was a low risk, 1.3–2.66 was an indeterminate risk and ≥2.67 was a high risk [26]. In the renal outcome analysis, the FIB-4 category at baseline and at 5 years, and changes from baseline to 5 years were used. The FIB-4 improved group was defined as a downgrade in the risk category, the FIB-4 stable group was defined as maintenance in an individual category, and the FIB-4 progressive group was defined as an upgrade in the risk category. Classification for chronic kidney disease (CKD) was made according to the KDIGO 2020 Clinical Practice Guideline [2].

### 2.4. Definitions of Diabetes and Diabetic Nephropathy

Diabetes was defined as fasting plasma glucose concentrations ≥126 mg/dL, random plasma glucose concentrations ≥200 mg/dL, and/or HbA1c values ≥ 6.5%, or being medicated for diabetes [27]. Diabetic nephropathy (DN) was defined according to the eGFR and severity of albuminuria and proteinuria [2,28]. At the first visit, eGFR and U-Alb/U-Cre ratios were measured, and the urinary dipstick test was performed on all patients. If proteinuria was identified with the dipstick test, the U-Pro/U-Cre ratio was additionally measured. The DN progression was defined as follows: category 1, an eGFR ≥ 30 mL/min/1.73 m^2^ and U-Alb/U-Cre ratio < 30 mg/g; category 2, an eGFR ≥ 30 mL/min/1.73 m^2^ and U-Alb/U-Cre ratio of 30–299 mg/g or U-Pro/U-Cre ratio of 0.15–0.49 mg/g; category 3, an eGFR ≥ 30 mL/min/1.73 m^2^ and U-Alb/U-Cre ratio ≥ 300 mg/g or U-Pro/U-Cre ratio ≥ 0.5 mg/g; category 4, an eGFR < 30 mL/min/1.73 m^2^ and any severity of albuminuria and proteinuria; and category 5, hemodialysis [28]. All enrolled patients had category 2 or category 3 DN in the present study. The patients were classified into the improved group, stable group, or progressive group according to the changes in DN category from baseline to 5 years.

### 2.5. Statistical Analysis

Continuous variables are shown as the median (lower and upper quartile). The percentage change in albuminuria and proteinuria was defined as the ΔU-Alb at any time point/U-Alb at baseline × 100 in patients with DN category 2 at baseline and the ΔU-Pro at any timepoint/U-Pro at baseline × 100 in patients with DN category 3 at baseline, respectively. The Dunn–Bonferroni test was used for the comparison of continuous variables between the groups. Fisher’s exact test with the Bonferroni correction was used for the comparison of categorical variables between the groups stratified by the DN outcome. The Kruskal-Wallis test was also performed for multiple comparison. The Wilcoxon signed-rank test was used for the comparison of paired continuous variables between the baseline data and data at any timepoint after baseline in individual groups stratified by the DN outcome. Repeated measures analysis of variance was used to compare the changes in individual parameters from baseline to 5 years among the groups stratified by the DN outcome. Spearman’s rank correlation coefficient was used for correlation analysis. A simple regression model analysis and multiple regression model analysis were performed to identify the factors associated with the ΔFIB-4 index. In the multivariate multiple regression analysis, variables in a univariate analysis were used for explanatory variables. In the renal outcome analysis, the initiation of dialysis and death from renal failure after 5 years from baseline was defined as an event. The Kaplan–Meier method was used to analyze time-to-event data, and the log-rank test was performed for comparison between the groups. Differences were considered significant when *p* was <0.05. All analyses were performed using IBM SPSS Statistics version 21 (IBM, Corp., Armonk, NY, USA).

## 3. Results

### 3.1. Characteristics of the Patients at Baseline According to Baseline DN Category and FIB-4 Index

The baseline characteristics of the patients are summarized according to the DN category at baseline (Table 1). There was no significant difference in age, sex, body weight, BMI, prevalence of dyslipidemia, and estimated salt intake among the groups. HbA1c values of category 2 were lower than category 3 (*p* < 0.05). Creatinine of category 2 was significantly lower than category 3 (*p* < 0.001). eGFR of category 2 was significantly higher than category 3 (*p* < 0.001). CKD stage was more progressive in category 3 compared to category 2 (*p* < 0.001). Regarding liver function, no significant difference was observed for AST, platelet count, and FIB-4 index. ALT of category 2 was significantly higher than category 3 (*p* < 0.05). The number of patients who received insulin injection therapy was lower in category 2 than in category 3 (*p* < 0.05). The baseline characteristics of the patients are summarized according to the FIB-4 index category at baseline (Appendix A). HbA1c values of the low-risk group were significantly higher than the other groups (*p* < 0.05). Creatinine of the indeterminate risk group was significantly lower than the other groups (*p* < 0.05). eGFR of the indeterminate risk group was significantly higher than the other groups (*p* < 0.05).

### 3.2. Characteristics of the Patients at Baseline According to DN Outcome

The baseline characteristics of the patients are summarized according to the DN category at 5 years (Table 2). There was no significant difference in age, sex, body weight, or BMI among the groups. Blood pressure and the prevalence of hypertension were not different among the groups. The prevalence of dyslipidemia in the improved group was significantly higher than that in the progressive group (*p* < 0.05). HbA1c values were not different among the groups. In the improved and stable groups, serum creatinine concentrations were lower and the eGFR was higher than those in the progressive group (*p* < 0.001), and these differences among the groups were statistically significant by Kruskal-Wallis test (*p* < 0.001). The prevalence of DN categories 2 and 3 was not different among the groups. There was no difference in liver enzymes, the platelet count or the FIB-4 index among the groups. Regarding pharmacological therapy, no patients received sodium-coupled glucose transporter 2 inhibitors in any of the groups. The number of patients who received insulin injection therapy was lower in the improved group than in the progressive group (*p* < 0.05). In all the parameters, there was no statistical difference between the improved group and stable group.

### 3.3. Characteristics of the Patients at 5 Years and Changes in the Parameters of the Groups

The characteristics of the patients at 5 years were compared among the groups and also compared with the baseline characteristics (Table 3). No significant differences among the groups at 5 years were observed for body weight, BMI, diastolic blood pressure, HbA1c, platelet count, AST, ALT, and estimated salt intake. No differences were observed for FIB-4. However, systolic blood pressure, U-Alb/U-Cre, and U-Pro/U-Cre were lower in the improved group, and U-Alb/U-Cre was also lower in the stable group compared to the progressive group (*p* < 0.05). Differences in eGFR, U-Alb/U-Cre, and U-Pro/U-Cre among the groups were statistically significant according to the Kruskal-Wallis test (*p* < 0.001). Compared to baseline, body weight, BMI, diastolic blood pressure, HbA1c, U-Alb/U-Cre, U-Pro/U-Cre, and AST decreased in the improved group; U-Pro/U-Cre decreased in the stable group; and body weight, U-Alb/U-Cre, and U-Pro/U-Cre increased in the progressive group. eGFR was significantly decreased from baseline in all groups (*p* < 0.001). According to the DN category, 29 patients with category 2 and 8 patients with category 3 at baseline improved to category 1 and category 2 in the improved group. In the progressive group, aggravation from category 2 to category 3 was observed in 32 patients, aggravation from category 2 to category 4 was observed in 4 patients and aggravation from category 3 to category 4 or category 5 was observed in 21 patients. Changes in BMI, HbA1c, eGFR, albuminuria and proteinuria, blood pressure, and salt intake of the groups from baseline to 5 years are shown in Figure 1. Changes in BMI (Figure 1A), eGFR (Figure 1C), and albuminuria and proteinuria (Figure 1D) were significantly different among the groups (*p* < 0.001). Changes were not significantly different among the groups in HbA1c (*p* = 0.279) (Figure 1B), in systolic blood pressure (*p* = 0.126) (Figure 1E), and in diastolic blood pressure (*p* = 0.110) (Figure 1F). Changes in salt intake were not different among the groups (*p* = 0.189) (Figure 1G).

### 3.4. Changes in the Parameters of the Groups

Changes in the parameters of the groups from baseline to 5 years are shown in Figure 1. BMI was significantly decreased in the improved group and was increased in the progressive group at 5 years compared with baseline (*p* < 0.001) (Figure 1A). HbA1c values were decreased in the improved group and maintained in the stable and progressive groups, but the changes were not significantly different among the groups (*p* = 0.279) (Figure 1B). The eGFR was significantly decreased at 5 years compared with baseline in all of the groups, and this decrease was more progressive in the progressive group than in the other groups (*p* < 0.001) (Figure 1C). Changes in albuminuria and proteinuria were significantly different among the groups (*p* < 0.001) (Figure 1D). Albuminuria and proteinuria were aggravated in the progressive and stable groups, while they were stable at the baseline level in the improved group. There was no change in systolic blood pressure from baseline to 5 years in any of the groups (*p* = 0.126) (Figure 1E). Diastolic blood pressure decreased from baseline to 5 years in the improved group, but this change was not significantly different between the groups (*p* = 0.110) (Figure 1F). Changes in salt intake from baseline were not significant in any of the groups, and changes were not different among the groups (*p* = 0.189) (Figure 1G).

### 3.5. Changes in ALT Concentrations and the FIB-4 Index

ALT concentrations were maintained at approximately the baseline level in the stable group from baseline to 5 years. ALT concentrations decreased in the improved and progressive group from baseline to 5 years, but this was not significant (Figure 2A). The FIB-4 index was maintained at approximately the baseline level at the various time points in the improved group, but it was increased from baseline to 5 years in the stable and progressive groups (Figure 2B). At 2 and 5 years, the FIB-4 index in the improved group was significantly lower than that in the progressive group (*p* = 0.049 and *p* = 0.026, respectively). There was no significant correlation between the ΔFIB-4 index and ΔeGFR (ρ = −0.022, *p* = 0.775) (Figure 2C). Weak but positive correlation between ΔFIB-4 index and albuminuria/proteinuria was observed (ρ = 0.22, *p* = 0.004) (Figure 2D).

### 3.6. Factors Associated with the ΔFIB-4 Index

To identify the changes in the factors that are associated with the ΔFIB-4 index, univariate and multivariate multiple regression analyses were performed (Table 3). In the univariate analysis, changes in albuminuria and proteinuria (*p* = 0.005) and systolic blood pressure (*p* = 0.039) were significantly associated with a change in the FIB-4 index. A change in the eGFR was not a significant factor. Changes in albuminuria and proteinuria were also associated with the ΔFIB-4 index by the multivariate analysis (*p* = 0.002). In the univariate regression analyses with baseline characteristics, systolic blood pressure and DN category were significantly associated with a change in the FIB-4 index (Appendix A). In the multivariate regression analysis, systolic blood pressure associated with a change in the FIB-4 index.

### 3.7. Renal Outcome and the FIB-4 Index

We next examined whether the FIB-4 index is associated with renal outcome of the patients. During the observation period (median observation period of 7.3 years), seven patients had dialysis or death from renal failure. The FIB-4 index at baseline (Figure 3A) and 5 years (Figure 3B) failed to stratify the prognosis. However, according to the change in the FIB-4 index category, the FIB-4 progressive group showed a significantly lower event-free survival rate than the FIB-4 improved group (*p* = 0.037) (Figure 3C).

## 4. Discussion

In the present longitudinal study, we investigated the interaction between changes in liver fibrosis in NAFLD and renal outcome of DN in patients with diabetes who received an educational course for DN. This study showed that changes in liver fibrosis evaluated by the FIB-4 index were associated with the outcome of DN and renal prognosis of patients with diabetes. Moreover, a change in albuminuria or proteinuria associated with changes in the FIB-4 index. These findings indicate a potential interaction between DN and hepatic fibrosis in patients with diabetes. 

Typical DN is characterized by the presence of albuminuria/proteinuria, followed by an impaired glomerular filtration rate (GFR) [1,2,3,29]. In this study, changes in the severity of DN stratified the changes in multiple parameters, such as BMI and HbA1c (Figure 1). Importantly, aggravation of the DN category was associated with progression of liver fibrosis (Figure 2B). This finding leads to the question, “which component of DN, albuminuria/proteinuria or the eGFR, is associated with changes in the FIB-4 index?”. The correlation analysis and regression analyses in this study showed that albuminuria/proteinuria, but not changes in the eGFR, were associated with changes in the FIB-4 index (Figure 2C,D and Table 4). Moreover, among the changes in multiple parameters, albuminuria/proteinuria was solely associated with changes in the FIB-4 index according to the multivariate analysis (Table 4). A recent study showed that the risk and phenotype of cardiovascular outcomes were different among phenotypes of diabetic kidney disease, such as albuminuria/proteinuria dominant type and impaired GFR dominant type [29]. Therefore, our results suggest a closer relationship between albuminuria/proteinuria and hepatic fibrosis in NAFLD than that between GFR and hepatic fibrosis in NAFLD. Additionally, a potential common background and effective intervention are assumed. To date, a specific treatment strategy for albuminuria/proteinuria independent of GFR has not been established. Indeed, multifactorial intervention, such as diet therapy, exercise, glycemic control, lowering blood pressure by a renin–angiotensin–aldosterone inhibitor, and lipid management, is recommended to prevent the progression of DN [1,2,3,29]. All of these individual interventions may be effective to prevent the progression of hepatic fibrosis in NAFLD [14,30,31]. In terms of clinical implementation, identifying common risk factors and developing an effective intervention for albuminuria/proteinuria and liver fibrosis in patients with NAFLD, diabetes, and DN are important. Body weight reduction and glycemic control influence the development of DN as well as liver fibrosis [1,2,14]. In the present study, changes in BMI and glycemic control were not the factors which affected the changes in FIB-4 index (Table 4). A possible explanation for this are the small changes in these parameters in the study cohort as shown in Figure 1A,B, which might not have been enough to affect the changes in liver fibrosis.

Accumulating evidence has indicated an association between the pathogenesis of NAFLD and the outcome of CVD [32,33]. However, the relationship of microvascular disease with NAFLD is inconclusive. According to epidemiological and cross-sectional studies in patients with type 2 diabetes, the prevalence of DN and microalbuminuria is higher in subjects with NAFLD than in those without NAFLD [34,35]. However, this finding was not found in other studies [36,37]. Yeung et al. reported that liver fibrosis measured with vibration-controlled transient elastography, but not liver steatosis measured with quantitative ultrasound techniques (controlled attenuation parameter), in patients with type 2 diabetes was associated with the risk of albuminuria [38]. Saito et al. conducted a longitudinal study including patients with type 2 diabetes without diabetic kidney disease and showed that a baseline FIB-4 index > 1.3 was associated with the onset of diabetic kidney disease and proteinuria [39]. These reports by Yeung et al. and Saito et al. support our finding that liver fibrosis in NAFLD is associated with albuminuria/proteinuria and DN. Our study, which evaluated parameters at baseline and 5 years, additionally showed that changes in the severity of albuminuria/proteinuria and DN were associated with changes in liver fibrosis in NAFLD.

The underlying mechanism to explain the interaction between the development of albuminuria/proteinuria and liver fibrosis remains unclear. Patients with DN and NAFLD have overlapping and concomitant conditions, such as hypertension, dyslipidemia, insulin resistance, and hyperglycemia. These conditions can enhance the renin–angiotensin system [40] and oxidative stress [41], and might simultaneously affect the development of albuminuria/proteinuria and liver fibrosis. Further study is required to determine if there is a causal relationship.

The prevalence of dyslipidemia was higher in the DN improved group than the stable group and progressive group (Table 2). The definition of dyslipidemia in this study was based on diagnosis in the medical record and prescriptions for dyslipidemia. Therefore, pharmacological treatments for dyslipidemia such as statins could be associated with DN outcome. Indeed, the inhibitory effect of statins on the progression of diabetic kidney disease was reported in a Chinese study [42]. The percentage of patients treated with insulin injections was lower in the improved group than in the progressive group, suggesting that patients in the improved group had an insulin-resistance state with more frequent dyslipidemia, rather than an insulin-dependent state.

According to the multivariate analysis in the present study, the change in albuminuria and proteinuria is the only factor that positively associates with the progression of liver fibrosis (Table 4). On the other hand, the baseline DN severity is not significant in the multivariate analysis (Appendix A). In terms of clinical implications, these data suggest that patients with prevention for the development of albuminuria and proteinuria might prevent the progression of liver fibrosis. Baseline systolic blood pressure negatively associates with the progression of liver fibrosis (Appendix A), and the change in systolic blood pressure, although it was significant only in univariate analysis, positively associates with the progression of liver fibrosis (Table 4), suggesting that patients with higher systolic blood pressure at baseline and a decrease in systolic blood pressure might contribute to attenuating the progression of liver fibrosis. In terms of clinical implementation, these data suggest that management of DN, regardless of the baseline DN severity, and blood pressure control might be effective in preventing the progression of liver fibrosis in DN patients with NAFLD. On the other hand, we could not identify the risk factors that aggravate liver fibrosis. Further investigation is required.

There are several limitations to this study. All patients were from a single center, and selection bias could have been present. The small sample size and the small number of patients with renal events are also a limitation. However, the longitudinal design and multiple detailed evaluations of clinical parameters should have compensated for the small sample size. The diagnosis of NAFLD and evaluation of liver fibrosis were not performed with a liver biopsy, but with noninvasive formula instead, which could affect the high prevalence of NAFLD in our study cohort. However, there are limitations of a liver biopsy as the gold standard for the diagnosis of NAFLD and liver fibrosis, such as invasiveness, a non-quantitative approach, and discordant pathological diagnosis among pathologists [43]. Therefore, a noninvasive approach such as elastography, which enables multiple evaluations, might be a possible procedure in the present longitudinal study. Changes in medication including pioglitazone, SGLT-2 inhibitors, and GLP-1 analogs could affect the disease progression of both NAFLD and DN. Because we collected medication data only at the baseline, any possible effect of changes in medication could not be investigated.

In conclusion, the outcome of DN, particularly changes in albuminuria/proteinuria, is associated with changes in liver fibrosis in patients with diabetes, NAFLD, and DN. Changes in liver fibrosis can be used to stratify the renal prognosis of these patients. A preventive and therapeutic approach to improve these pathogeneses could be optimized in the future.

## Figures and Tables

**Figure 1 nutrients-15-03248-f001:**
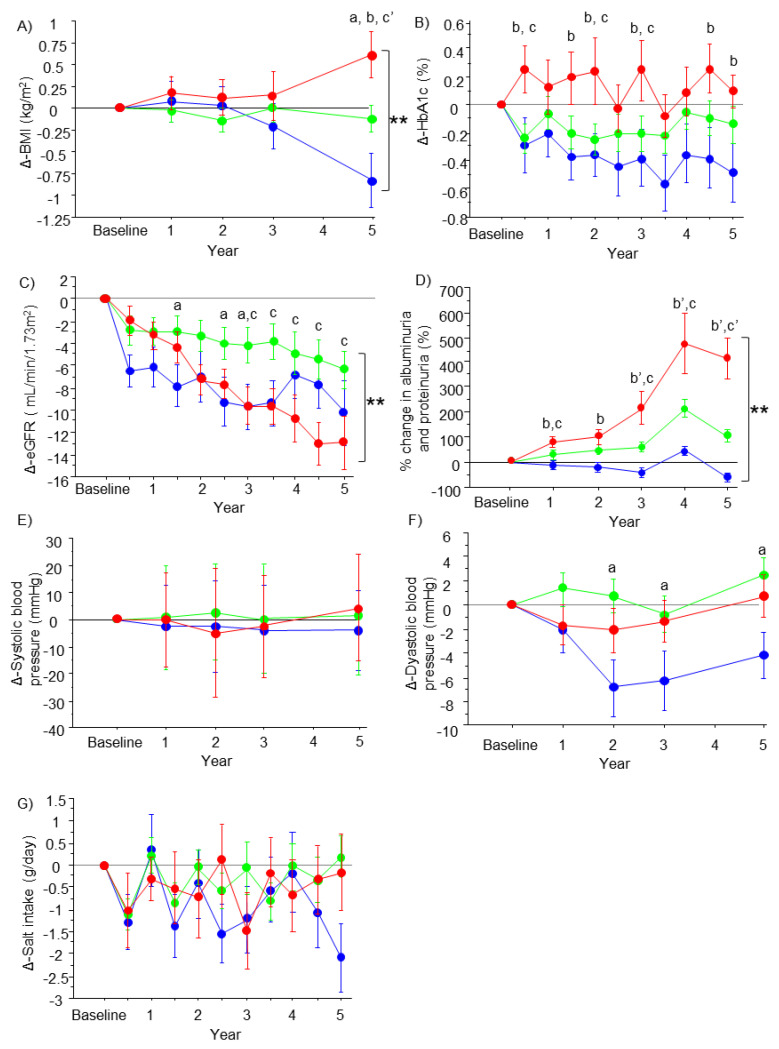
Changes in the parameters in the groups. The line graphs represent the changes in BMI (**A**), HbA1c (**B**), eGFR (**C**), % change in albuminuria and proteinuria (**D**), systolic blood pressure (**E**), diastolic blood pressure (**F**), and salt intake (**G**) from baseline. Dots and error bars represent the mean value and standard deviation (blue, improved group; green, stable group; and red, progressive group). Significant differences between the improved and stable groups (a or a’), between the improved and progressive groups (b or b’), and between the stable and progressive groups (c or c’) are shown as *p* < 0.05 or *p* < 0.001 and were analyzed by the Dunn–Bonferroni test. ** *p* < 0.001 indicates significant difference among the groups by repeated measures analysis of variance.

**Figure 2 nutrients-15-03248-f002:**
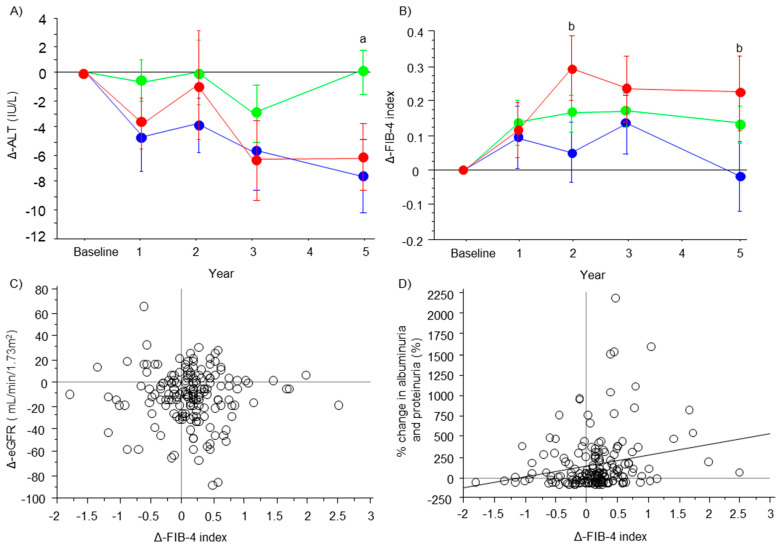
Changes in ALT concentrations and the FIB-4 index. The line graphs represent changes in ALT concentrations (**A**) and the FIB-4 index (**B**) from baseline. Dots and error bars represent the mean value and standard deviation (blue, improved group; green, stable group; and red, progressive group). Significant differences between the improved and stable groups (a), between the improved and progressive groups (b), are shown as *p* < 0.05 or *p* < 0.001 and were analyzed by the Dunn–Bonferroni test. The scatter plot graphs show the correlation between changes in the FIB-4 index and changes in the eGFR (**C**) or % change in albuminuria and proteinuria (**D**).

**Figure 3 nutrients-15-03248-f003:**
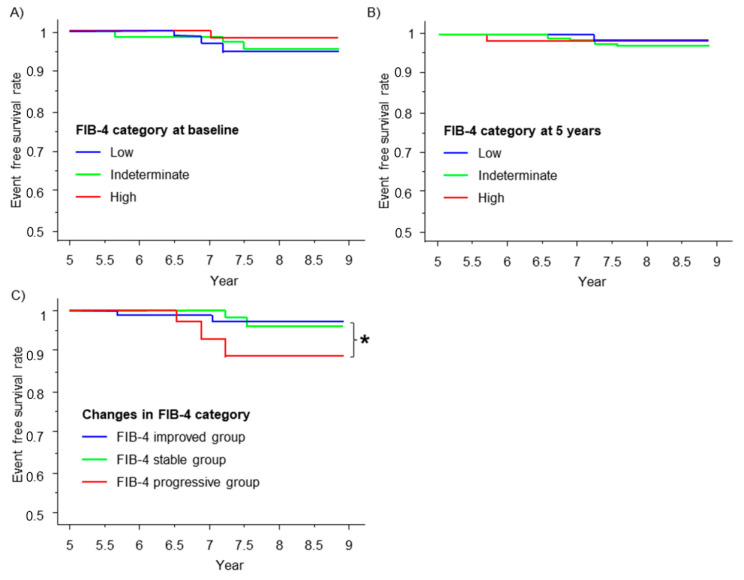
Renal outcome and the FIB-4 index. The renal event-free survival curve is shown using the Kaplan–Meier method stratified by the fibrosis risk category (blue line, green line, and red line represent low risk, indeterminate risk, and high risk, respectively) according to the FIB-4 index at baseline (**A**) and at 5 years (**B**). Renal event-free survival curve stratified by the changes in risk category (blue line, green line, and red line represent the FIB-4 improved group, FIB-4 stable group, and FIB-4 progressive group, respectively) from baseline to 5 years (**C**). * *p* < 0.05; the log-rank test was used to compare the FIB-4 progressive group and the FIB-4 improved group.

**Table 1 nutrients-15-03248-t001:** Baseline characteristics of the patients according to baseline DN category.

	DN Category	
	Category 2 n = 127	Category 3 n = 62
Age, years	64 (52–76)	63 (53–73)
Female, n (%)	40 (31.5)	20 (32.3)
Body weight, kg	65.7 (51.4–80)	68.9 (56.1–81.7)
BMI, kg/m^2^	25.7 (21.8–29.6)	26 (21.1–30.9)
Systolic blood pressure, mmHg	133 (113–153)	137 (123–150)
Diastolic blood pressure, mmHg	78 (66–90)	77 (57–97)
Hypertension, n (%)	68 (53.5) *	46 (74.2)
Dyslipidemia, n (%)	43 (33.9)	17 (27.4)
HbA1c, %	7 (5.6–8.4) *	7.5 (6.3–8.8)
Creatinine, mg/dL	0.84 (0.53–1.15) **	1.03 (0.59–1.47)
eGFR, mL/min/1.73 m^2^	67.1 (43.1–91.1) **	52 (30.5–73.5)
CKD stage G1/G2/G3a/G3b, n (%)	15/67/37/8 (11.8/52.8/29.1/6.3) **	4/21/13/24 (6.5/33.9/21/38.7)
U-Alb/U-Cre, mg/g	79.4 (42.0–116.8)	-
U-Pro/U-Cre, mg/g	-	1.3 (0.7–0.9)
Estimated salt intake, g/day	10.6 (7.3–13.9)	10.9 (8.2–13.6)
AST, U/L	22 (13–31)	20 (11–29)
ALT, U/L	24 (7–41) *	20 (8–32)
Platelet count, ×10^3^/µL	221 (157–285)	224 (139–309)
FIB-4 index	1.27 (0.52–2.01)	1.24 (0.42–2.06)
SGLT2 inhibitor, n (%)	0	0
GLP-1 agonist, n (%)	11 (8.7)	5 (8.1)
DPP-4 inhibitor, n (%)	61 (48)	35 (56.5)
Pioglitazone, n (%)	10 (7.9)	2 (3.2)
Insulin injection, n (%)	47 (37) *	34 (54.8)

The U-Alb/U-Cre ratio was measured in patients with DN category 2. The U-Pro/U-Cre ratio was measured in patients with DN category 3. The Dunn–Bonferroni test was used for the comparison of continuous variables between the groups. Fisher’s exact test with the Bonferroni correction was used for the comparison of categorical variables between the groups. Abbreviations: BMI—body mass index; HbA1c—hemoglobin A1c; eGFR—estimated glomerular filtration rate; CKD—chronic kidney disease; U-Alb/U-Cre—urine albumin to urine creatinine ratio; U-Pro/U-Cre—urine protein to urine creatine ratio; DN—diabetic nephropathy; AST—aspartate aminotransferase; ALT—alanine aminotransferase; FIB-4—fibrosis 4; SGLT2—sodium glucose cotransporter 2; GLP-1—glucagon-like peptide-1; DPP4—dipeptidyl peptidase 4. * *p* < 0.05 and ** *p* < 0.001, compared with category 3.

**Table 2 nutrients-15-03248-t002:** Baseline characteristics of the patients according to DN outcome.

	DN Outcome		
	Improved Group n = 37	Stable Group n = 95	Progressive Group n = 57
Age, years	65 (53—78)	64 (51–77)	64 (54–75)
Female, n (%)	14 (37.8)	31 (32.6)	16 (28.1)
Body weight, kg	66.7 (49.2–84.2)	66.2 (51.7–80.7)	67 (53.2–80.8)
BMI, kg/m^2^	26.9 (22.9–30.9)	25.7 (22.1–29.3)	25.5 (21.0–29.9)
Systolic blood pressure, mmHg	136 (114–158)	132 (116–148)	136 (118–154)
Diastolic blood pressure, mmHg	79 (65–93)	75 (60–90)	76 (64–88)
Hypertension, n (%)	23 (62.2)	56 (58.9)	43 (75.4)
Dyslipidemia, n (%)	17 (45.9) *	31 (32.6)	12 (21.1)
HbA1c, %	7.3 (6.0–8.6)	7.2 (5.9–8.6)	7.1 (5.7–8.5)
Creatinine, mg/dL ^(AA)^	0.78 (0.49–1.07) **	0.85 (0.56–1.14) **	1.04 (0.62–1.46)
eGFR, mL/min/1.73m^2 (AA)^	73 (52.8–93.3) **	64 (40.8–87.3) **	51.5 (26.5–76.5)
U-Alb/U-Cre, mg/g	66.4 (40.9–91.9) **	78.1 (33.8–122.4) *	112.7 (43.0–182.4)
U-Pro/U-Cre, mg/g ^(AA)^	0.7 (0.3–1.1) *	1.2 (0.68–1.73) **	2.4 (0.88–3.93)
DN category, category 2/category 3, (%)	29/8 (78.4/21.6)	62/33 (65.3/34.7)	36/21 (64.2/36.8)
Estimated salt intake, g/day	11.6 (8.6–14.6)	10.5 (7.6–13.4)	10.7 (7.3–14.1)
AST, U/L	21 (12–31)	22 (14–30)	19 (10–28)
ALT, U/L	18 (13–41)	24 (11–38)	21 (8–34)
Platelet count, ×10^3^/µL	229 (150–310)	221 (159–284)	223 (151–295)
FIB-4 index	1.23 (0.58–1.89)	1.27 (0.41–2.12)	1.24 (0.54–1.93)
SGLT2 inhibitor, n (%)	0 (0)	0 (0)	0 (0)
GLP-1 agonist, n (%)	2 (5.4)	7 (7.4)	7 (12.3)
DPP-4 inhibitor, n (%)	17 (45.9)	54 (56.8)	25 (43.9)
Pioglitazone, n (%)	4 (10.8)	4 (4.2)	4 (7.0)
Insulin injection, n (%)	10 (27.0) *	39 (41.1)	32 (56.1)

The U-Alb/U-Cre ratio was measured in patients with DN category 2. The U-Pro/U-Cre ratio was measured in patients with DN category 3. The Dunn–Bonferroni test was used for the comparison of continuous variables between the groups. Fisher’s exact test with the Bonferroni correction was used for the comparison of categorical variables between the groups. Abbreviations: BMI—body mass index; HbA1c—hemoglobin A1c; eGFR—estimated glomerular filtration rate; U-Alb/U-Cre—urine albumin to urine creatinine ratio; U-Pro/U-Cre—urine protein to urine creatine ratio; DN—diabetic nephropathy; AST—aspartate aminotransferase; ALT—alanine aminotransferase; FIB-4—fibrosis 4; SGLT2—sodium glucose cotransporter 2; GLP-1—glucagon-like peptide-1; DPP4—dipeptidyl peptidase 4. * *p* < 0.05 and ** *p* < 0.001, compared with the progressive group ^(AA)^ *p* < 0.001 by Kruskal-Wallis test.

**Table 3 nutrients-15-03248-t003:** Characteristics of the patients at 5 years.

	DN Outcome		
	Improved Group n = 37	Stable Group n = 95	Progressive Group n = 57
Body weight, kg	65.9 (51.3–80.6) ^§^	66.8 (51.5–82.1)	69.1 (54.4–83.8) ^§^
BMI, kg/m^2^	25.4 (20.8–30.0) ^§^	25.5 (20.7–30.2)	25.225 (20.2–30.2)
Systolic blood pressure, mmHg	131 (107–155) *	135 (113–157)	139 (117–162)
Diastolic blood pressure, mmHg	72 (59–85) ^§^	75 (58–92)	77 (61–94)
HbA1c, %	6.9 (6.0–7.9) ^§^	7.1 (6.1–8.2)	7.2 (6.2–8.3)
eGFR, mL/min/1.73m^2 (AA)^	63 (30.8–95.3) ^§§^	60 (34–86) ^§§^	41 (9–73) ^§§^
U-Alb/U-Cre, mg/g ^(AA)^	18.5 (7.8–29.2) **^§§^	119 (8.2–229.8) **	475.9 (197.8–754) ^§§^
U-Pro/U-Cre, mg/g ^(A)^	0.4 (0.275–0.525) *^§^	3.45 (1.25–8.15) ^§§^	2.45 (1.35–8.25) ^§^
DN category, 1/2/3/4/5	29/8/0/0/0	0/62/33/0/0	0/0/32/21/4
Estimated salt intake, g/day	10 (7.4–12.6) ^§^	10.7 (7.0–14.4)	9.5 (5.4–13.6)
AST, U/L ^(A)^	19 (12–26) ^§^	21.5 (13–31)	20 (13–27)
ALT, U/L ^(A)^	19 (10–28) ^†§^	21 (6–36)	17 (4–30) ^§^
Platelet count, ×10^3^/µL	221 (140–302)	209 (137–281)	219 (151–287)
FIB-4 index	1.30 (0.45–2.14)	1.48 (0.63–2.32) ^§^	1.57 (0.73–2.42) ^§^

The U-Alb/U-Cre ratio was measured in patients with DN category 2. The U-Pro/U-Cre ratio was measured in patients with DN category 3. The Dunn–Bonferroni test was used for the comparison of continuous variables between the groups. Abbreviations: BMI–body mass index; HbA1c–hemoglobin A1c; eGFR—estimated glomerular filtration rate; U-Alb/U-Cre—urine albumin to urine creatinine ratio; U-Pro/U-Cre—urine protein to urine creatine ratio; DN—diabetic nephropathy; AST—aspartate aminotransferase; ALT—alanine aminotransferase; FIB-4—fibrosis 4. * *p* < 0.05 and ** *p* < 0.001, compared with the progressive group; ^†^ *p* < 0.05, compared with the stable group; ^§^ *p* < 0.05 and ^§§^ *p* < 0.001, compared with baseline data. ^(A)^ *p* < 0.05 and ^(AA)^ *p* < 0.001 by Kruskal-Wallis test.

**Table 4 nutrients-15-03248-t004:** Regression analysis of changes in the FIB-4 index.

	Explanatory Variable	Partial Regression Coefficient (×10^−3^)	*t* Value	*p* Value	95% C.I
Univariate					
	% change in albuminuria and proteinuria	0.35	2.84	0.005	0.0001–0.001
	Δ-eGFR	−1.0	−0.45	0.656	−0.007–0.004
	Δ-HbA1c	54.0	1.43	0.154	−0.021–0.129
	Δ-Salt intake	−7.0	−0.72	0.47	−0.027–0.013
	Δ-BMI	1.0	0.05	0.958	−0.047–0.05
	Δ-Systolic blood pressure	4.0	2.08	0.039	0.0002–0.008
	Δ-Diastolic blood pressure	4.0	1.43	0.15	−0.002–0.011
Multivariate					
	% change in albuminuria and proteinuria	0.31	2.409	0.017	0.0001–0.001
	Δ-Systolic blood pressure	2.0	1.167	0.245	−0.002–0.007

A simple regression model analysis and multiple regression model analysis. Abbreviations: BMI—body mass index; HbA1c—hemoglobin A1c; eGFR—estimated glomerular filtration rate.

## Data Availability

The data that support the findings of this study are available on request from the corresponding author, H.T. The data are not publicly available due to their containing information that could compromise the privacy of research participants.

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
