# Peer review of "Change in Liver Fibrosis Associates with Progress of Diabetic Nephropathy in Patients with Nonalcoholic Fatty Liver Disease"

_nutrients, 2023, doi:10.3390/nu15143248_

Round 1

Reviewer 1 Report

The study performed by Terasaka and colleagues evaluates the interaction between changes in liver fibrosis and renal outcomes in patients with NAFLD and DN. The idea is interesting but the study needs more clarity.

In the background:

- the authors explain that glycemic control impacts on NAFLD outcomes and that a comprehensive care is important. What does it mean a comprehensive care? what about the importance of body weight on the prognosis of NAFLD patients? Up to know body weigh reduction is the most effective treatment to avoid fibrosis progression. In addition, several DM drugs have an impact not only on controlling glicemia but also on body weight. Please, expand. 

- The first hypothesis of the study is relevant and is proven along the study. Nonetheless, the second part of the hypothesis has not been tested in this specific study. 

- In relation to this, the aims of the study have to be reformulated. This is not a study were the authors compare the impact of a multidisciplinary care and education program on liver fibrosis. There is no a control group without the education program to evaluate this aim. 

In the methods:

- How were the patients included in an education course? High educational level? consecutive? please, specify. 

- Were patients without missing data and alcohol consumption excluded? how was alcohol intake evaluated? 

- 91.3% of patients with DM and DN presented NAFLD defined by HSI>30? Selection bias? Please, comment on this on the discussion compared with previous studies. 

- In the statistical analysis all methods used are mentioned but there is no explanation for what specific purpose except for the multivariable analysis (were there is a spelling mistake). Please, define what are the main outcomes of the study, at what time points and the analysis used according to the specific aims.

Results:

- There should be a real baseline characteristics of the study cohort according to KD stage at baseline.

- Is could be interesting to see differences at baseline according to fibrosis stage as well (maybe as supplementary).

- Why dyslipidemia is higher in the improved group than in the others? the reader would have expected the opposite. Please, discuss.

- tables should include statistical analysis used on the legend.

- characteristics of the patients at 5 years and changes in the parameters of the groups are repeating the same information twice. Please, organize it in one section.

- Were there changes in medication along time that could have influence disease progression? This is a very important point that has not been described in the study.

- What about insulin differences between groups? Is there a higher effect of insulin compared with other treatments on DN progression? Please comment. 

- There seems to be an effect of several parameters such as body weight or BMI, blood pressure or HbA1c on DN progression. These factors are known to influence fibrosis progression in NAFLD, please comment on the possible interaction of these factors on the fibrosis groups. Is there a reduction in BMI leading to an improved FIB-4 group at 5 years? 

- In figure 1 there are missing data at year 4, is this a mistake? please, correct. 

- there should be a mistake in figure 2B where significance is on year 2 instead of year 3. 

- The correlation between FIB-4 change and albuminuria-proteinuria is very low. 

- What is the impact of other related variables on the change of FIB-4? age is included as a variable in the FIB-4 formula, and differences include 5 years time frame, is there a real influence of age on the risk of progression? What about dylipidemia, sex or DN stage? Is there an influence of DM treatments on the progression of NAFLD fibrosis? 

- Some additional analysis could add extra value information to the study. What is the high risk group of patients with DN that will have fibrosis progression? This is important given that this would be the group of patients that need more attention and interventional strategies. 

Discussion

- with the results of this study the authors can not claim that a change in albuminuria or proteinuria affects liver fibrosis. There has not been a causality relationship. Please, correct. 

- the impact of the study on clinical practice has to be highlighted based on the results. 

English needs a major revision throughout the entire manuscript. 

Author Response

Reviewer 1

Comments and Suggestions for Authors

The study performed by Terasaka and colleagues evaluates the interaction between changes in liver fibrosis and renal outcomes in patients with NAFLD and DN. The idea is interesting but the study needs more clarity.

Thank you so much for positive impression and valuable suggestions. We revised the manuscript according to the comments.

In the background:

R1-1

- the authors explain that glycemic control impacts on NAFLD outcomes and that a comprehensive care is important. What does it mean a comprehensive care? what about the importance of body weight on the prognosis of NAFLD patients? Up to know body weight reduction is the most effective treatment to avoid fibrosis progression. In addition, several DM drugs have an impact not only on controlling glycemia but also on body weight. Please, expand.

As reviewer suggested, we expanded the description about “comprehensive care” for NAFLD (page 1, line 33 – 35) and described the effect of body weight reduction and pharmacological therapy for NAFLD (page 1 - 2, line 45 - line 47).

R1-2

- The first hypothesis of the study is relevant and is proven along the study. Nonetheless, the second part of the hypothesis has not been tested in this specific study.

- In relation to this, the aims of the study have to be reformulated. This is not a study were the authors compare the impact of a multidisciplinary care and education program on liver fibrosis. There is no a control group without the education program to evaluate this aim.

We agree with the reviewer. The second hypothesis was removed from the manuscript (page 2, line 52 - 53).

In the methods:

R1-3

- How were the patients included in an education course? High educational level? consecutive? please, specify.

Regarding inclusion for educational course, we revised the manuscript as follow. “All the patients with diabetes and DN who visited Heiwadai Hospital from September 2013 to December 2015 were consecutively recruited to an education course.” (page 2, line 60 - 63). Unfortunately, information about educational level of the individual patients were not recorded. Educational course was consecutive and detail information of the course was explained in the following paragraph.

R1-4

- Were patients without missing data and alcohol consumption excluded? how was alcohol intake evaluated?

Information about alcohol intake were based on the interview by physician and nurse in charge. There was no missing data for alcohol consumption and 6 patients were excluded accordingly as shown in Figure S1.

R1-5

- 91.3% of patients with DM and DN presented NAFLD defined by HSI>30? Selection bias? Please, comment on this on the discussion compared with previous studies.

So far, we could not find the epidemiological study that identified the prevalence of NAFLD in the patients with DM and DN. According to the meta-analysis by Younossi ZM et al (J Hepatol 2019, 71, 793-801), approximately 55% of the type 2 diabetes patients have NAFLD. According to our recent meta-analysis, prevalence of NAFLD in type 2 diabetes patients was approximately 60% (Cho EL et al. Gut 2023. Accepted). According to the study by Saito H et al (Sci Rep 2021, 11, 11753), approximately 50-60% of type 2 diabetes patients who suspected to have NAFLD developed to DN in 12 months. These evidences suggest that prevalence of NAFLD in the patients with DN could be estimated as at least 60% or higher. On the other hand, diagnosis for NAFLD using HSI could affect the relatively high prevalence of the present study. According to Lee JH et al (Dig Liver Dis 2010, 42, 503-508), sensitivity and specificity of HSI ≥ 30 is 92.5% for NAFLD diagnosis, that could overestimate the prevalence of NAFLD in our cohort. We add discussion regarding this point as study limitation (page 13, line 415 - 417).

R1-6

- In the statistical analysis all methods used are mentioned but there is no explanation for what specific purpose except for the multivariable analysis (were there is a spelling mistake). Please, define what are the main outcomes of the study, at what time points and the analysis used according to the specific aims.

We add the purpose of statistics used in the study (page 3, line 138 - 146).

Results:

R1-7

- There should be a real baseline characteristics of the study cohort according to DKD stage at baseline.

Thank you so much for your suggestion. We add the table (Table 1) and description in the result section (page 4, line 156 - 166). 

R1-8

- It could be interesting to see differences at baseline according to fibrosis stage as well (maybe as supplementary).

Thank you so much for your suggestion. We add the table as supplementary table 1 and description in the result section (page 4, line 166 - 171) 

R1-9

- Why dyslipidemia is higher in the improved group than in the others? the reader would have expected the opposite. Please, discuss.

The reason was unclear but we would like to discuss that point. The definition of dyslipidemia was according to the diagnosis in medical record and prescription for dyslipidemia. Therefore, pharmacological treatment for dyslipidemia such as statin could associate the DN outcome. Indeed, inhibitory effect of statins on progression of diabetic kidney disease was reported in Chinese study (Zou S et al. CMAJ. 2023). The percentage of the patients treated with insulin injection was fewer in the improved group than progressive group, suggesting that patients in the improved group were insulin resistance state with more frequent dyslipidemia rather than insulin dependent state. We discussed this point in the discussion section (page 12, line 387 - 395).

R1-10

- tables should include statistical analysis used on the legend.

Statistical procedure used in the table was added on the individual legends.

R1-11

- characteristics of the patients at 5 years and changes in the parameters of the groups are repeating the same information twice. Please, organize it in one section.

We combined the two sections into one (page 6, line 213 - 237 and page 7 line 250 - 266) .

R1-12

- Were there changes in medication along time that could have influence disease progression? This is a very important point that has not been described in the study.

As reviewer suggested, changes in medication including pioglitazone, SGLT-2 inhibitor and GLP-1 analog could affect the disease progression of both NAFLD and DN. Unfortunately, because we collected the data of medication only at the baseline, possible effect of changes in medication could not investigated. This is very important point as reviewer suggested and described in study limitation (page 13, line 421 - 424). 

R1-13

- What about insulin differences between groups? Is there a higher effect of insulin compared with other treatments on DN progression? Please comment.

Causal relationship between the insulin injection and DN progression cannot be assessed in the present study. So far, there was no evidence and consensus that insulin injection aggravate DN. In the present study, insulin injection represent the pathogenesis of insulin dependent state with relatively developed-disease course in diabetes that might induce progression in DN.   

R1-14

- There seems to be an effect of several parameters such as body weight or BMI, blood pressure or HbA1c on DN progression. These factors are known to influence fibrosis progression in NAFLD, please comment on the possible interaction of these factors on the fibrosis groups. Is there a reduction in BMI leading to an improved FIB-4 group at 5 years?

The answer for the question from the reviewer could be found in regression model analysis in Table 4. Changes in BMI and glycemic control were not the factors which affect the changes in FIB-4 index. As reviewer suggested, these factors generally affect the development of liver fibrosis. Possible explanation was small changes in these parameters in the study cohort as shown in Figure 1A and 1B, that might not be enough to affect the changes in liver fibrosis. We added discussion regarding this point (page 12. line 358 - 363). There was no significant differences in the changes in BMI and HbA1c according to the outcome of FIB-4 index (improved/stable/progressive; -0.08/-0.18/-0.06 for BMI, -0.2/-0.2/-0.35 for HbA1c) (data not shown).

R1-15

- In figure 1 there are missing data at year 4, is this a mistake? please, correct.

We apologize for misleading data presentation. Several parameters were not collected at 4 years. Time point for the collection of individual data was added in the manuscript (page 3, line 100 - 104) and we modified the graph in Figure 1 and Figure 2.

R1-16

- there should be a mistake in figure 2B where significance is on year 2 instead of year 3.

Thank you so much for your notification. Description in the result section was wrong and Figure 2B was correct. The error in the text was corrected (page 9, line 281).

R1-17

- The correlation between FIB-4 change and albuminuria-proteinuria is very low.

We modified the description (page 9, line 284).

R1-18

- What is the impact of other related variables on the change of FIB-4? age is included as a variable in the FIB-4 formula, and differences include 5 years time frame, is there a real influence of age on the risk of progression? What about dyslipidemia, sex or DN stage? Is there an influence of DM treatments on the progression of NAFLD fibrosis?

All the patient aged by 5 years. Therefore, it is statistically impossible to include the variable for the analysis of regression model and such parameter statistically cannot be a significant factor for delta-FIB-4 index with the different changes among the patients in 5 years. On the other hand, as the reviewer suggested, baseline age and other parameters could be analyzed. Data (supplementary table 2) and related-description (page 10, line 302 - 306 and page 12 - 13, line 396 - 410) were added.  

R1-19

- Some additional analysis could add extra value information to the study. What is the high risk group of patients with DN that will have fibrosis progression? This is important given that this would be the group of patients that need more attention and interventional strategies.

Thank you so much for the valuable comment. Similar to the response to the comment above, we analyzed the baseline characteristics to identify the factors which affect the changes in liver fibrosis (supplementary table 2).

Discussion:

R1-20

- with the results of this study the authors can not claim that a change in albuminuria or proteinuria affects liver fibrosis. There has not been a causality relationship. Please, correct.

According to the suggestion from reviewer, “affect” was replaced by “associate” (page 11, line 330).

R1-21

- the impact of the study on clinical practice has to be highlighted based on the results.

Thank you so much for the comment. We added discussion (page 10, line 302 - 306 and page 12 - 13, line 396 - 410).

Reviewer 2 Report

Change in liver fibrosis associates with outcome of diabetic nephropathy in patients wiith diabetes and nonalcoholic fatty liver disease.

Terasaka Y et al.

The study enrolled a small cohort of 189 subjects with diabetes, diabetic nephropathy, and nonalcoholic fatty liver disease. The objective is to explore the association of liver fibrosis, assessed by the fibrosis-4 (FIB-4) index, and kidney parameters in subjects in which NAFLD was defined according to the Hepatic Steatosis Index (HSI). Changes in FIB-4 were positively associated with changes in albuminuria. Consistently, over a median observation period of 7.3 years, FIB-4 progression was associated with a lower event-free survival rate for a composite outcome of dialysis or death from renal failure.

My observations follow:

Title:

I suggest changing as: “Change in liver fibrosis associates with changes in kidney parameters in patients with diabetic nephropathy and nonalcoholic fatty liver disease.”

Abstract:

Rows 16-18: this sentence is unclear. Please, rephrase it.

Methods:

Rows 59-62: please rephrase this sentence: “who?”

Row 65: I was unable to find Figure S1; please provide it.

Rows 73-74: please, define renal failure (i.e., eGFR <15 ml/min/1.73 m2)

Rows 117-122: The definition of DN stages as reported by Authors (using eGFR threshold of 30 ml/min/1.73 m2 for stage 1 and stage 2) is misleading, it could be more appropriate to use, for example, “strata” or “categories”.

Results:

Rows 169-183. This entire section is very hard to follow.

I suggest chancing as follows: “No significant differences between groups at 5 years were observed for body weight, BMI, diastolic blood pressure, HbA1c, eGFR, platelet count, AST, ALT, and estimated salt intake. No differences were observed for FIB-4. Otherwise, systolic blood pressure, U-Alb/U-Cre, and U-Pro/U-Cre were lower in the improved group and U-Alb/U-Cre also in the stable group as compared with the progressive group.

Compared to baseline, body weight, BMI, diastolic blood pressure, HbA1c, U-Alb/U-Cre, and U-Pro/U-Cre, and AST decreased in the improved group; U-Pro/U-Cre decreased in the stable group; body weight, U-Alb/U-Cre, and U-Pro/U-Cre increased in the progressive group. eGFR was significantly decreased from baseline in all groups at baseline (p<0.001).”

Looking at “Characteristics of the patients at 5 years” and “Changes in the parameters of the groups” paragraphs, many results seem to be reported twice. Please, try to simplify.

For both Table 1 and table 2 it could be of interest to include p-value for ANOVA or Kruskal-Wallis tests.

Furthermore, it could be of interest to describe (as supplementary material) baseline characteristics according to DN “stages” at baseline, also providing information about eGFR categories in both DN “stages”.

Renal outcome was defined as initiation of dialysis and death from renal failure after 5 years from baseline in the Statistical Analysis (page 3; row 139). Median observation period has been reported as 7.3 years in the Results section (page 8; row 258): however, in Figure 3, observation seems to start at year 5, with an observation of up to 4 years (overall 9 years from baseline). Please reconcile these apparent discrepancies.

Only seven patients had dialysis or death from renal failure. Is it possible to define a wider composite renal outcome including patients with a reduction in eGFR of 40% or 50% or 57% (i.e., doubling of serum creatine)? This might allow to increase the number of incident events.

Discussion:

In “Methods”, Authors report that all patients enrolled underwent the education course for DN. The conclusion that a “successful education … regarding DN could be effective to prevent the progression of hepatic fibrosis in NAFLD and to improve the renal outcome of DN” is not supported by evidence considering that there is not a control group (no educational intervention).

References:

From reference 21 onwards the numerical order reported in the text seems to be wrong. Please correct. Consistently, ref 41 appears in the text, but is not reported in the references list.

The manuscript is not easy to read, and the English needs much improvement.

Author Response

Reviewer 2

The study enrolled a small cohort of 189 subjects with diabetes, diabetic nephropathy, and nonalcoholic fatty liver disease. The objective is to explore the association of liver fibrosis, assessed by the fibrosis-4 (FIB-4) index, and kidney parameters in subjects in which NAFLD was defined according to the Hepatic Steatosis Index (HSI). Changes in FIB-4 were positively associated with changes in albuminuria. Consistently, over a median observation period of 7.3 years, FIB-4 progression was associated with a lower event-free survival rate for a composite outcome of dialysis or death from renal failure.

My observations follow:

Title:

R2-1

I suggest changing as: “Change in liver fibrosis associates with changes in kidney parameters in patients with diabetic nephropathy and nonalcoholic fatty liver disease.”

Thank you so much for the suggestion. As reviewer suggested, the word of “outcome” in the title is inappropriate because our study is not interventional trial. On the other hand, “kidney parameters” might be ambiguous. Also, our study includes the data of association between liver fibrosis and renal prognosis. Probably the reviewer considered the “confusing” classification of DN “stage” in our study (actually, it was according to the classification of Japan Diabetes Society. Later, according to the suggestion of the reviewer, “DN stage” was modified to “DN categories”), the reviewer avoided to use “stage” here. Taken together, we would like to modify the title as follow if it is OK with the reviewer: “Change in liver fibrosis associates with progress of diabetic nephropathy in patients with nonalcoholic fatty liver disease.”

Abstract:

R2-2

Rows 16-18: this sentence is unclear. Please, rephrase it.

We modified the sentence accordingly (page 1, line 15 - 18).

Methods:

R2-3

Rows 59-62: please rephrase this sentence: “who?”

We modified the sentence accordingly (page 2, line 60 - 63).

R2-4

Row 65: I was unable to find Figure S1; please provide it.

We apologize for our mistake. Figure S1 was correctly submitted the supplementary files for revision.

R2-5

Rows 73-74: please, define renal failure (i.e., eGFR <15 ml/min/1.73 m2)

As the reviewer suggested, definition for renal failure was missing and we added it in the document (page 2, line 75 - 76).

R2-6

Rows 117-122: The definition of DN stages as reported by Authors (using eGFR threshold of 30 ml/min/1.73 m2 for stage 1 and stage 2) is misleading, it could be more appropriate to use, for example, “strata” or “categories”.

As reviewer suggested, we replace “stage” with “category” for evaluation of DN progression.

Results:

R2-7

Rows 169-183. This entire section is very hard to follow.

I suggest chancing as follows: “No significant differences between groups at 5 years were observed for body weight, BMI, diastolic blood pressure, HbA1c, eGFR, platelet count, AST, ALT, and estimated salt intake. No differences were observed for FIB-4. Otherwise, systolic blood pressure, U-Alb/U-Cre, and U-Pro/U-Cre were lower in the improved group and U-Alb/U-Cre also in the stable group as compared with the progressive group. Compared to baseline, body weight, BMI, diastolic blood pressure, HbA1c, U-Alb/U-Cre, and U-Pro/U-Cre, and AST decreased in the improved group; U-Pro/U-Cre decreased in the stable group; body weight, U-Alb/U-Cre, and U-Pro/U-Cre increased in the progressive group. eGFR was significantly decreased from baseline in all groups at baseline (p<0.001).”

Thank you so much for the suggestion. We simplified the description of the section (page 6, line 215 - 226).

R2-8

Looking at “Characteristics of the patients at 5 years” and “Changes in the parameters of the groups” paragraphs, many results seem to be reported twice. Please, try to simplify.

We combined the two sections into one (page 6, line 213 - 237 and page 7 line 250 - 266) .

R2-9

For both Table 1 and table 2 it could be of interest to include p-value for ANOVA or Kruskal-Wallis tests.

Kruskal-Wallis test was added in the tables and document (page 3, line 141; page 5, line 193 - 194 and page 6, line 220 - 222).

R2-10

Furthermore, it could be of interest to describe (as supplementary material) baseline characteristics according to DN “stages” at baseline, also providing information about eGFR categories in both DN “stages”.

Thank you so much for the comment. Similar suggestion was provided by another reviewer and we generated the new table (Table 1) and information about CKD stage and related information were added in the table (page 3, line 115 – 116; page 4, line 156 - 171 and Table 1).

R2-11

Renal outcome was defined as initiation of dialysis and death from renal failure after 5 years from baseline in the Statistical Analysis (page 3; row 139). Median observation period has been reported as 7.3 years in the Results section (page 8; row 258): however, in Figure 3, observation seems to start at year 5, with an observation of up to 4 years (overall 9 years from baseline). Please reconcile these apparent discrepancies.

Thank you so much for the comment. We double check the data and the median observation period and Kaplan–Meier curves in Figure 3 were correct. In the study including the cohort with different observation period, length (period) of Kaplan–Meier curve is affected by the subjects with the longest observation period. In the present study, subjects with approximately 9 year-observation period were included in all the FIB-4 index categories.    

R2-12

Only seven patients had dialysis or death from renal failure. Is it possible to define a wider composite renal outcome including patients with a reduction in eGFR of 40% or 50% or 57% (i.e., doubling of serum creatine)? This might allow to increase the number of incident events.

That’s interesting suggestion to be investigated. Unfortunately, our data collection after 5 year was limited and information except event is missing. Small number of the patients with event are obvious study limitation, which should be described in the manuscript (page 13, line 412 - 413).

Discussion:

R2-13

In “Methods”, Authors report that all patients enrolled underwent the education course for DN. The conclusion that a “successful education … regarding DN could be effective to prevent the progression of hepatic fibrosis in NAFLD and to improve the renal outcome of DN” is not supported by evidence considering that there is not a control group (no educational intervention).

We agree with the reviewer. The second hypothesis in the method and related conclusion in discussion were removed from the manuscript (page 11, line 332 - 334).

References:

R2-14

From reference 21 onwards the numerical order reported in the text seems to be wrong. Please correct. Consistently, 41 appears in the text, but is not reported in the references list.

We found the error in the reference numbering and corrected it.
